# Clinical Management of Pathogen-Negative Tuberculous Meningitis in Adults: A Series Case Study

**DOI:** 10.3390/jcm11216250

**Published:** 2022-10-23

**Authors:** Yuqin He, Yanzhu Huang, Di Wu, Yingying Wu, Minghuan Wang

**Affiliations:** 1Department of Neurology, Tongji Hospital, Tongji Medical College, Huazhong University of Science and Technology, Wuhan 430030, China; 2Department and Institute of Infectious Disease, Tongji Hospital, Tongji Medical College, Huazhong University of Science and Technology, Wuhan 430030, China; 3Department of Oncology, Tongji Hospital, Tongji Medical College, Huazhong University of Science and Technology, Wuhan 430030, China

**Keywords:** tuberculous meningitis, pathogen-negative, management, diagnosis

## Abstract

Tuberculosis remains a serious world public health problem. Tuberculous meningitis (TBM) is the one of most severe forms of extrapulmonary tuberculosis. However, the insensitivity and time-consuming requirement of culturing the pathogen *Mycobacterium tuberculosis*, the traditional “gold standard” diagnostic test for TBM, often delays timely diagnosis and treatment, resulting in high disability and mortality rates. In our series case study, we present five pathogen-negative TBM cases who received empirical anti-tuberculosis therapy with a good clinical outcome. We describe in detail the clinical symptoms, laboratory test results, and imaging findings of the five patients from symptom onset to dynamic follow-up. We then summarize the similarities of the clinical characteristics of the presented patients, as well as shared features in laboratory and imaging tests, and proceed to analyze the challenges in the timely diagnosis of TBM. Finally, we argue that monitoring of cerebrospinal fluid markers and imaging are critical for the diagnosis and treatment of TBM, and emphasize the importance of differential diagnosis in cases when tuberculous meningitis is highly suspected despite negative findings for that etiology.

## 1. Introduction

Tuberculosis is among the oldest documented and historically most lethal infectious diseases of humans, and remains a serious world public health problem. Tuberculosis meningitis (TBM) is the most severe form of tuberculosis, with a high mortality rate and risk of neurological involvement. The global burden of TBM is far underestimated because of the difficulties in making a correct diagnosis [1]. The traditional “gold standard” diagnostic test for TBM involves growing the pathogen *Mycobacterium tuberculosis* in culture, a process that is insensitive and time-consuming, thus often resulting in delayed diagnosis and treatment. The main reasons for a false-negative result of *M. tuberculosis* culture are low abundance of the pathogen in cerebrospinal fluid (CSF), and the need for a high standard of laboratory equipment and procedures [2]. As an alternative to the traditional Ziehl–Neelsen smear (ZN smear), emerging Gene-xpert/RIF (Xpert) and metagenomic next-generation sequencing (mNGS) methods are more sensitive and faster, but the negative results still cannot exclude the diagnosis of TBM. Furthermore, the volume, preservation, processing, and antibiotic treatment of CSF can affect test results [3]. Therefore, in geographic regions with a high burden of TBM, especially in countries with limited laboratory access, the diagnosis of TBM often depends on clinical features and the treatment decisions are made empirically, based on an integrated assessment rather than objective evidence of the pathogen. Timely and accurate initiation of empirical anti-tuberculosis treatment can significantly improve mortality and prognosis. Diagnostic guidelines of TBM proposed by the British Infection Society [4], Indian [5], Chinese medical association [6], and the American Thoracic Society [7] indicate that all patients suspected of TBM should start empirical anti-tuberculosis treatment immediately with a four-drug regimen, without waiting for microbiological or molecular diagnostic confirmation. However, no unified clinical diagnostic standard is so far available, and the available scoring systems that are widely used in the clinical setting are notoriously insensitive and nonspecific. Hence, we now present a clinical report of five cases of pathogen-negative TBM who received empirical antituberculosis therapy in conjunction with close clinical management and monitoring, resulting in each case in a good outcome. We hope that this summary of our experience supports practical suggestions for an improved differential diagnosis and optimal management of the disease.

## 2. Materials and Methods

### 2.1. Patient Recruitment

We conducted a retrospective study from April 2021 to August 2021 in Tongji Hospital (including the main hospital area, Sino-French New City branch, and Optical Valley Branch), Tongji Medical College, Huazhong University of Science and Technology, Wuhan, Hubei Province, China. All patients (*n* = 5) were suspected of TBM, and improved after receiving empirical anti-tuberculosis treatment, as established by monitoring for at least three months of follow-up.

### 2.2. Inclusion Criteria

(1) Age ≥ 18; (2) TBM was first diagnosed in our hospital; (3) received empirical antituberculosis therapy; (4) follow-up time ≥ 3 months.

### 2.3. Exclusion Criteria

(1) Age < 18; (2) TBM has been diagnosed at other hospitals; (3) start of antituberculosis treatment after a positive *M. tuberculosis* culture or positive smear.

### 2.4. Data Collection

We collected information on demographics (age, gender, and personal history), exposure history of tuberculosis, history of smoking, alcohol, or other substance use, and history of other illnesses (such as tuberculosis and diabetes, hepatitis B, HIV/AIDS, syphilis, immune-mediated rheumatic disease, and organ transplantation), symptom duration (the interval from onset of symptoms to hospital admission), ATT time (time from the onset of symptoms to the start of anti-tuberculosis treatment), prodrome, clinical symptoms (headache, fever, night sweat, hyponatremia, cranial nerve palsy symptoms, altered consciousness, and stiff-neck), laboratory results (*M. tuberculosis* culture, ZN smear, mNGS, Xpert, bacterial culture, bacterial staining, Indian ink staining, virus antibody of CSF specimen, fungal antigen, virus antibody, mNGS, T-SPOT, and antinuclear antibody in blood), evidence of extracranial tuberculosis (chest-CT and spinal cord-MRI), CSF findings upon repeated lumbar puncture after admission (pressure, clearance, glucose, chloride, lactate dehydrogenase activity, lactate, immunoglobulin G, IgM, and IgA) and changes in head-MRI imaging after admission.

## 3. Results

Five previously healthy patients, whose main symptoms were recurrent fever and headache lasting at least 10 days, were referred to our hospital after a first course of antibiotic treatment was ineffective. The patients had nonspecific accompanying symptoms such as a cold, cough, fatigue, anorexia, and muscle soreness. Two patients had hyponatremia. One patient had cranial nerve palsy symptoms presenting with intractable hiccup. Four patients had alteration of consciousness, one patient presented with significant psychiatric symptoms and cognitive impairment manifesting as rage and aggressive behaviors and spatiotemporal disorientation, one patient had recurrent epileptic seizures manifesting as limb convulsions and loss of consciousness, and two patients showed sleepiness. The “stiff-neck” sign of meningeal irritation appeared in five patients. The diagnosis assays for TBM included CSF sampling with *M. tuberculosis* culture, ZN smear, and Xpert in CSF, which were negative for all five patients. One patient was positive for human *M. tuberculosis* in CSF using mNGS. The T-SPOT assays of four patients were positive. Except for *M. tuberculosis*, tests for pathogen antigens or antibodies were negative in CSF and blood samples of the five patients. The CSF samples of two patients were examined for antibodies related to autoimmune encephalitis, with negative results. There were no chest-CT imaging features indicative of tuberculous infection in the five patients. Four patients also underwent cervical spine MRI examination, with no evidence of spinal tuberculosis and tuberculous myelitis. The detailed clinical information, laboratory test results, and imaging findings for the five patients are shown in Table 1.

Based on consideration of clinical manifestations, CSF laboratory results, and imaging findings, four of the five patients received standard empirical anti-tuberculosis treatment with 600 mg isoniazid, 450 mg rifampicin, 1500 mg pyrazinamide, and 750 mg ethambutol (HRZE) with dexamethasone used to inhibit inflammatory reaction after the first lumbar puncture. Due to suffering an epileptiform seizure, the remaining patient underwent the first lumbar puncture on the fifth day after admission, with initiation of empirical anti-tuberculosis treatment as above, starting after the second lumbar puncture on the eighteenth day after admission.

During the course of HRZE treatment, the patients underwent lumbar puncture and brain MRI imaging at various time points to monitor any changes in condition and treatment response. The dynamic changes in various CSF indices in the five patients are shown in Figure 1. Three of the five patients showed a fluctuating downtrend in the white cell count during the first month of antibiotic treatment. Lymphocytes remained the predominant white cell type throughout the course of treatment in all five patients. At the last lumbar puncture, the CSF indicators returned to the normal range after treatment. Nonetheless, imaging findings deteriorated in four of the five patients during the first treatment month. In particular, the numbers and size of intracranial lesions in three patients grew for a time but had shrunk or disappeared at the last brain MRI examination. One patient had three brain MRI examinations within one month of symptom onset, which revealed worsening bilateral (but predominantly left side) meningeal enhancement in conjunction with frontal lobe edema. Five patients had enhancement of leptomeninges, and one patient had significant enhancement of the pontine cistern as well the cerebellar vermis cistern. There were no findings of hydrocephalus in four patients during the sequential imaging examination. Figure 2 presents the imaging results of three patients before and after treatment.

The percentage of monocytes/lymphocytes/neutrophils (right Y axis); the total white cell count (WCC; 106/L), and the concentrations of glucose (mmol/L × 10^−2^)/chloride (mmol/L)/protein (mg/L), lactate dehydrogenase(U/L), lactate (mmol/L × 10^−2^), and the concentrations of IgG//IgM//IgA in CSF (left Y axis); IgG//IgM//IgA (mg/L);

Sequence of lumbar puncture (X axis); when the total number of leukocytes is less than 50 × 10^6^/L, the cells are not classified; see Appendix A for details of lumbar puncture time and various index values.

Abbreviations: CSF—cerebrospinal fluid; TBM—tuberculous meningitis; IgG—immunoglobulin G; IgM—immunoglobulin M; IgA—immunoglobulin A.

## 4. Discussion

The clinical symptoms of our five cases all began with recurrent fever lasting for more than ten days, with obvious meningeal irritation signs. Their CSF samples consistently showed lymphocytic-predominant pleocytosis, elevated protein, and low glucose. Two of the five patients had brain parenchymal nodules on imaging, which might represent infectious lesions or, more likely, inflammatory granuloma. Their diagnosis of tuberculous meningitis was not confirmed by traditional methods, including CSF culture, smear, and the currently recommended Gene Xpert method. An infection caused by fungi, other bacteria, viruses, parasites, and other pathogens were further excluded through laboratory examination of the CSF and blood samples. Considering the above clinical characteristics, we immediately started empirical anti-tuberculosis treatment with the standard antibiotic polypharmacy. Their diagnosis of TBM was confirmed to be correct through the composite of clinical symptoms, repeated lumbar puncture findings, and MR/CT imaging examinations. Follow-up of our series of patients showed a good clinical outcome, without neurological deficits. Our results favor a diagnosis of *M. tuberculosis* in patients with unexplained persistent fever and headache, along with obvious meningeal irritation signs, upon exclusion of other pathogens and central nervous system diseases.

Clinicians often use scoring systems based on clinical manifestations, cerebrospinal fluid characteristics, and imaging to differentiate TBM from other central nervous system infectious diseases. The most commonly used scores are the Lancet consensus scoring system [8] and Thwaites’ system [9]. According to the Lancet consensus scoring system, two patients were classified as probable and the remaining three as possible, while according to Thwaites’ system, all five of our patients met the diagnostic criteria of TBM. Detailed scores are shown in Appendix A. Consistent with previous clinical studies, we find that the Lancet consensus scoring system has high specificity and low sensitivity, whereas the converse is true for Thwaites’ system [10]. The duration of symptoms is the strongest predictor for differentiating TBM from viral meningitis and bacterial meningitis [11]. Thus, symptoms duration of two–four weeks raises a suspicion of TBM rather than bacterial meningitis [12]. Other studies have shown that hyponatremia and altered consciousness are strong predictors of present TBM, while also predicting poor prognosis [13]. The diagnostic significance of the proportion of lymphocytes and neutrophils in CSF remains a matter of controversy. Most studies have shown that the majority of CSF nucleated cells are lymphocytes, which is more indicative of TBM [14,15,16,17,18]. A meta-analysis found that more than four-fifths of TBM patients had low CSF/blood glucose ratio [19], which we likewise observed in the present study group. However, it is a pity that adenosine deaminase (ADA) in serum and CSF of patients was not detected in our study. ADA is an indirect predictor of tuberculosis and it has been used for the diagnosis of the pleural, peritoneal, and pericardial forms of tuberculosis [20]. Especially in pleural fluid, ADA has shown a high overall diagnostic sensitivity, specificity, positive likelihood ratio (PLR), and negative likelihood ratio (NLR) for TB [21]. Tuon et al. (2010) reported that CSF adenosine deaminase activity determination can be of benefit as a rule-in or rule-out test when values of less than 4 U/L and greater than 8 U/L [22] are recorded. However, the clinical application of ADA is limited due to the existence of the blood–brain barrier and the small number of cerebrospinal fluid samples. ADA can also be found in patients infected with HIV or who have other HIV-associated neurological diseases, such as cryptococcal meningitis, lymphomatous meningitis, and cytomegalovirus disease [23]. According to the unified clinical standard of Lancet, 2010 [8] and the guideline for central nervous system tuberculosis of the Chinese Academy of Neurology in 2019 [6], the ADA test was not included in the diagnostic standard, so we did not detect the ADA value in serum and CSF.

Within one month of anti-tuberculosis treatment, the symptoms of five of our patients had improved, even though the imaging manifestations had deteriorated in three patients, a phenomenon known as a paradoxical reaction [24]. Recent studies have shown that the paradoxical reaction may be a normal manifestation of anti-tuberculosis treatment, and has nothing to do with the prognosis and morbidity [25]. We note that the deterioration of imaging is more like a delayed manifestation of the disease. Regarding neurological signs, one of our patients had an intractable hiccup and another had mild cognitive impairment at the time of admission; culprit lesions of the medulla oblongata and left frontal lobe respectively appeared on a head MRI one month later. Overall, hydrocephalus is the most common complication of TBM, occurring in 40% of cases according to the largest follow-up study conducted by Mohammad Wasay [25], but this was not a finding in our cases.

We reviewed 16 cases of pathogenic negative tuberculous meningitis since 2020 [26,27,28,29,30,31,32,33,34,35,36,37,38,39,40,41] (see Appendix A). These case reports are similar to our cases, mainly because of their atypical clinical manifestations or lesion sites, such as one-and-a-half syndrome [35], nonconvulsive status epilepticus [36], oculomotor palsy [38], lesions involving clivus [31], and midbrain [32]. Although there was no etiological diagnostic evidence, clinicians highly suspected tuberculous meningitis based on the general symptoms of tuberculosis, CSF examination, and imaging findings, and began empirical ATT treatment, which ultimately led to a good clinical outcome for most patients.

Tuberculosis treatment guidelines in the People’s Republic of China call for the completion of a course of empirical treatment once started, unless another alternative etiology emerges [6]. Therefore, there is a need for comprehensive differentiation and diagnosis of clinically suspected pathogen-negative cases, in order to achieve timely and appropriate anti-TBM treatment. If a patient’s clinical symptoms continue to deteriorate, or deteriorate again after initial improvement within one month after the start of empirical anti-tuberculosis treatment, physicians must consider the possibilities of paradoxical reaction and misdiagnosis. Therefore, blood and CSF indicators should be monitored at least every seven days during the first month of anti-tuberculosis treatment; subsequently, the patient can be followed up once a month through a lumbar puncture, blood tests, and imaging examination [42]. Previous case reports have shown that infectious meningitis, rheumatic immune diseases, and tumors can be misdiagnosed as TBM. For example, the clinical symptoms of fungal meningitis [43], neurobrucellosis [44], and leptomeningeal leukemia [45] can appear as recurrent bouts of fever along with headache. The cellular population in CSF in these diseases is typically composed of lymphocytes, with additional findings of decreased glucose and raised protein, which constitutes a pattern that is difficult to distinguish from TBM. Many forms of autoimmune meningitis such as GFAP encephalitis, and AQP4 encephalitis [46] can mimic the features of intracranial infection such as TBM, but are naturally unresponsive to antibiotic or antiviral treatment. Such patients may have masking of their true condition because of the bactericidal effect of anti-tuberculosis treatment and co-treatment with steroids, which calls for monitoring of early changes in condition upon treatment.

We note the limitation that our study is a single-center retrospective study, which could have resulted in selection bias and recall bias in data collection. There were only a few cases, which does not support strong qualitative or quantitative conclusions.

## 5. Conclusions

When patients with recurrent fever and headache leading to suspicion of TBM are not responsive to empirical antibiotic treatment, there is a need for careful anamnesis, with the recording of all relevant personal history, sojourn history, and symptoms of various bodily systems. Differential diagnosis calls for screening against rheumatism, autoimmune encephalitis-related antibodies, fungal, bacterial, and viral antigens, or antibodies, as well as markers of parasitic infection. As soon as possible after eliminating a range of alternate diagnoses, patients with high suspicion of TBM infection should start empirical anti-tuberculosis treatment; timely and appropriate therapy and follow-up disease monitoring are critical for obtaining good outcomes.

## Figures and Tables

**Figure 1 jcm-11-06250-f001:**
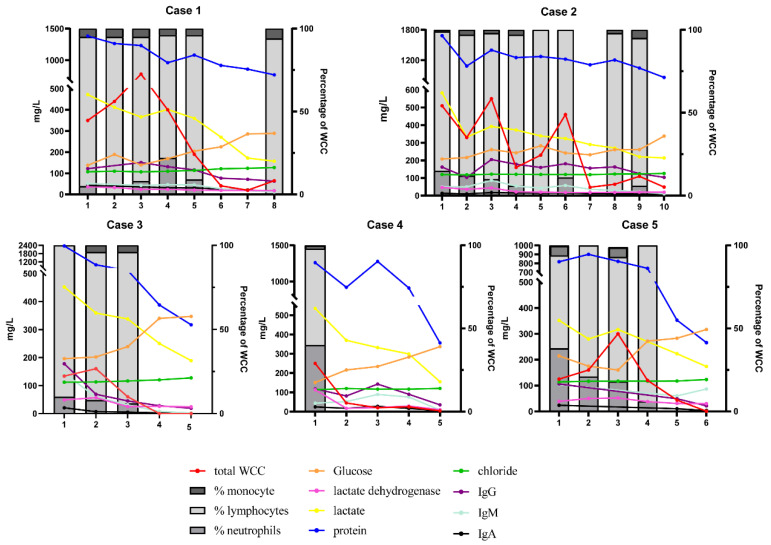
Dynamic changes in various CSF indices in five patients with TBM.

**Figure 2 jcm-11-06250-f002:**
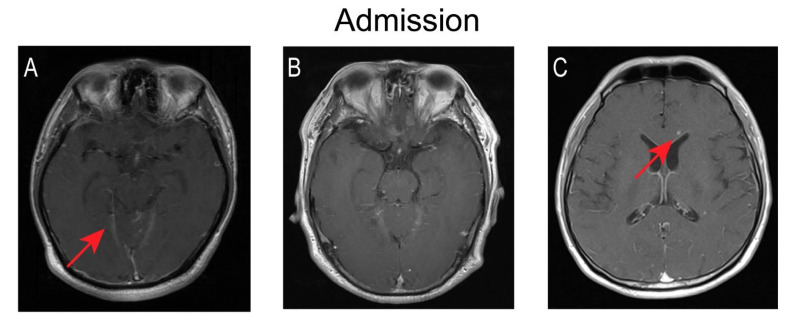
Changes of brain MRI before and after anti-tuberculosis treatment. The MRI sequence is Gd-enhanced MRI T1 cross-sectional; (**A**–**C**) The images of cases 1, 2, and 4 at admission. (**A**) Significant enhancement of pontine cistern as well as cerebellar vermis cistern. (**B**) No abnormal findings. (**C**) Scattered nodules of varying sizes in the brain parenchyma. (**D**–**F**) The images of cases 1, 2, and 4 after treatment for one month. (**D**) New nodular enhancement lesions were seen in the medulla oblongata in one patient. (**E**) Multiple new nodular enhancement lesions can be seen in the left frontal lobe of another case. (**F**) The lesions in the lateral ventricular angle were larger and more intense. (**G**–**I**) The images of cases 1, 2, and 3 after treatment for eight, three, and six months respectively. Follow-up imaging upon treatment showed resolution. (**G**) The lesions in the medulla oblongata of case 1. (**H**) The lesions in the left frontal lobe of case 2. (**I**) The lesions in the lateral ventricular angle of case 4.

**Table 1 jcm-11-06250-t001:** The detailed clinical information and laboratory test results of the five TBM patients.

Variables			Case 1	Case 2	Case 3	Case 4	Case 5
Age			27	56	48	40	62
Sex			Male	Male	Male	Male	Female
Potentially relevant personal history			Eats seafood	Neighbors have pigeons	Breakfast shop owner	Construction worker	Retiree
Exposure history of tuberculosis			Yes	No	No	No	No
Illness history			No	No	No	No	No
History of Smoke/Drink/Drug			No	No	No	No	No
Symptom duration (days)			29	14	11	19	12
Prodrome			Cold and cough	Cold and anorexia	Toothache and muscle soreness and anorexia	Diarrhea and fatigue	Cold
Clinical syndrome		Headache	Yes	Yes	Yes	Yes	Yes
		Fever	Yes	Yes	Yes	Yes	Yes
		Night sweat	Yes	No	Yes	No	No
		Hyponatremia	Yes	No	Yes	No	Yes
		Cranial nerve palsy symptoms	intractable hiccup	No	No	No	No
		Altered consciousness	Lethargy	Psychiatric symptoms and cognitive impairment	Lethargy	No	Epileptic seizures
		Stiff-Neck	Yes	Yes	Yes	Yes	Yes
Laboratory results	CSF Specimen	*M. tb* culture	−	−	−	−	-
		ZN smear	−	−	−	−	-
		mNGS	−	−	−	*Human M.tb*	-
		Xpert	−	−	−	−	-
		Bacterial culture	−	−	−	−	-
		Bacterial staining	−	−	−	−	-
		India ink staining	−	−	−	−	-
		Virus antibody *	−	/	−	−	-
		Ab related to AE *	−	/	/	/	-
		WCC * (10^6^/L)	350	510	134	250	125
		Protein * (mg/L)	1383	1683	2356	1260	819
		Glucose * (mmol/L)	1.38	2.08	1.96	1.51	2.15
	Blood specimen	Fungal antigen *	+	−	−	−	-
		Virus antibody *	−	/	−	−	-
		mNGS	−	−	−	−	-
		T-SPOT	+	+	+	+	-
		ANA	−	/	/	/	/
Evidence of extracranial tuberculosis		Chest-CT	-	−	−	−	−
		Cervical Spinal cord MRI	-	−	−	/	−
ATT time(days)			30	15	12	20	31
Follow up time(months)			8	3	6	6	3
Neurological sequelae			No	No	No	No	No

Legend: Virus antibody *—IgG and IgM antibodies of ECHO, PVB19, EVB, CA16, CVB, MV, VZV, CMV, RV, TOX, HSV I, HSV II; Ab related to AE *—antibodies related to autoimmune encephalitis, include GFAPR Ab, LGI-1, NMDAR, AMPAR1 Ab, AMPAR2 Ab, GABAR Ab, CASPR2 Ab; WCC *—White blood cell count value in CSF at the first lumbar puncture; Protein *—protein value in CSF at the first lumbar puncture; Glucose *—glucose value in CSF at the first lumbar puncture; Fungal antigen *—antigen of 1-3 beta d glucan and Aspergillus galactomannan; “-”—negative; “+”—positive; “/”—No performed in this patient. Abbreviations: CSF—cerebrospinal fluid; *Mycobacterium tuberculosis*—*M. tuberculosis/M. tb;* ZN smear—Ziehl–Neelsen smear; Xpert—Gene-xpert/RIF; mNGS—metagenomic next-generation sequencing; ATT—anti-tuberculosis treatment; IgG—immunoglobulin G; IgM—immunoglobulin M; IgA—immunoglobulin A; HRZE—isoniazid, rifampicin, pyrazinamide, ethambutol; Ab—antibodies; AE—autoimmune encephalitis; GFAPR Ab—glial fibrillary acidic protein receptor antibody; LGI-1 Ab—leucine-rich glioma inactivated 1 antibody; NMDAR—N-methyl-D-aspartate receptor antibody; AMPAR1 Ab—α-amino-3-hydroxy-5-methyl-4-isoxazolepropionic acid receptor antibody, type 1; AMPAR2 Ab—α-amino-3-hydroxy-5-methyl-4-isoxazolepropionic acid receptor antibody, type 2; GABAR Ab—gamma-amino butyric acid receptor antibody; CASPR2 Ab—contactin-associated pro-tein-like 2 antibody.

## Data Availability

Not applicable.

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
