# Peer review of "Clinical Management of Pathogen-Negative Tuberculous Meningitis in Adults: A Series Case Study"

_jcm, 2022, doi:10.3390/jcm11216250_

Round 1

Reviewer 1 Report

Comments to the author:

Major comment:

1) Consider describing ADA value in the CSF of the patients and discussing the degree of usefulness to diagnose TGM.

Minor comment:

2) Correct minor grammar errors (examples: an article, a comma, a hyphen, a space, spelling, Etc.).

Author Response

1) Consider describing ADA value in the CSF of the patients and discussing the degree of usefulness to diagnose TGM.

Response:Thanks for your suggestion. We have described more details in the discussion as follows:

“However, it is a pity that adenosine deaminase(ADA) in serum and CSF of patients were not detected in our study. ADA is an indirect predictor of tuberculosis and it has been used for the diagnosis of the pleural, peritoneal and pericardial forms of tuberculosis [1]. Especially in pleural fluid, ADA with a high overall diagnostic sensitivity, specificity, positive likelihood ratio (PLR) and negative likelihood ratio (NLR) for TB [2].Tuon et al, 2010 reported that CSF adenosine deaminase activity determination can be of benefit as a rule-in or rule-out test when values of less than 4 U/L and greater than 8 U/L[3]. However, the clinical application of ADA is limited due to the existence of blood brain barrier and small amount of cerebrospinal fluid samples. ADA can also be found in patients infected with HIV or who have other HIV-associated neurological diseases, such as cryptococcal meningitis, lymphomatous meningitis, and cytomegalovirus disease[4]. According to the unified clinical standard of lancet, 2010 [5] and the guideline for central nervous system tuberculosis of the Chinese Academy of Neurology in 2019[6], ADA test was not included in the diagnostic standard, so we did not detect the ADA value in serum and CSF.” (page 7, line 43-52; page 8, line1-5 in Revised-track version)

2) Correct minor grammar errors (examples: an article, a comma, a hyphen, a space, spelling, Etc.).

Response: We have carefully read through the manuscript and extensively edited the expression of the whole article.

Reference

  1. Segura, R. M.; Pascual, C.;  Ocaña, I.;  Martínez-Vázquez, J. M.;  Ribera, E.;  Ruiz, I.; Pelegrí, M. D. Adenosine deaminase in body fluids: A useful diagnostic tool in tuberculosis. Clinical Biochemistry 1989, 22, 141-148.
  2. Shaw, J. A.; Diacon, A. H.; Koegelenberg, C. F. N. Tuberculous pleural effusion. Respirology 2019, 24, 962-971.
  3. Tuon, F. F.; Higashino, H. R.;  Lopes, M. I. B. F.;  Litvoc, M. N.;  Atomiya, A. N.;  Antonangelo, L.; Leite, O. M. Adenosine deaminase and tuberculous meningitis—A systematic review with meta-analysis. Scandinavian Journal of Infectious Diseases 2010, 42, 198-207.
  4. Corral, I.; Quereda, C.;  Navas, E.;  Martín-Dávila, P.;  Pérez-Elías, M. J.;  Casado, J. L.;  Pintado, V.;  Cobo, J.;  Pallarés, E.;  Rubí, J.; et al. Adenosine deaminase activity in cerebrospinal fluid of HIV-infected patients: limited value for diagnosis of tuberculous meningitis. European Journal of Clinical Microbiology and Infectious Diseases 2004, 23, 471-476.
  5. Marais, S.; Thwaites, G.;  Schoeman, J. F.;  Török, M. E.;  Misra, U. K.;  Prasad, K.;  Donald, P. R.;  Wilkinson, R. J.; Marais, B. J. Tuberculous meningitis: a uniform case definition for use in clinical research. The Lancet Infectious Diseases 2010, 10, 803-812.
  6. Tuberculous Meningitis Professional Committee, T. B., Chinese Medical Association. 2019 Chinese guidelines for the diagnosis and treatment of central nervous system tuberculosis. Chinese Journal of Infectious Diseases 2020, 38, 400-408.

Reviewer 2 Report

There are many challenges of managing tuberculosis meningitis (TBM), including the inefficiency of pathogen culturing techniques, which can result in delayed diagnosis and treatment and poor outcomes.  In this report, He et al. describe five patients who tested negative for M. Tuberculosis, yet were treated with anti-Tuberculosis therapy regardless of absence of pathogen detection with good clinical outcome and argue that monitoring of cerebrospinal fluid markers and imaging are critical when tuberculosis meningitis is suspected, even if M. Tuberculosis is not detected.  Further, the authors suggest that a diagnosis of M. tuberculosis can be made in patients with unexplained persistent fever headache and signs of irritation, upon exclusion of other pathogens and central nervous system diseases. This is a clearly presented study that effectively argues for proactive treatment that is impactful for patients for TBM and does not rely upon culture of M. Tuberculosis

Minor comments:

Abstract—last sentence is written incorrectly.  Instead of “when patients are highly suspicion of” to when tuberculosis meningitis is highly s

Author Response

Minor comments:

Abstract—last sentence is written incorrectly. Instead of “when patients are highly suspicion of” to when tuberculosis meningitis is highly s

Response: We have revised the last sentence of abstract as follow “Finally, we argue that monitoring of cerebrospinal fluid markers and imaging are critical for the diagnosis and treatment of TBM, and emphasize the importance of differential diagnosis in cases when tuberculous meningitis is highly suspicion despite negative findings for that etiology”. (page 1, line 29-30 in Revised-track version)

Reviewer 3 Report

Extensive editing of the English language and style is required.

Also, it would be interesting if you would add a table in the Discussion chapter presenting a comparison between your cases and other cases presented in articles on the same subject from 2020-2021-2022.

Author Response

1) Extensive editing of the English language and style is requiredï¼›

Response: We have extensively edited the expression of the whole article.

2) It would be interesting if you would add a table in the Discussion chapter presenting a comparison between your cases and other cases presented in articles on the same subject from 2020-2021-2022.

Response: We have already added a table in the Discussion chapter presenting a comparison between our cases and other cases presented in articles on the same subject from 2020-2021-2022 (See in Supplement Table 4).

Reviewer 4 Report

This is a very good presentation of a series of patients suffering from tuberculous meningitis (TBM), all from ther same hospital in Wuhan, China, but in whom the identification of the causative organism failed and who had to be treated on an empirical basis. The outcome was good in all cases, demonstrating that the therapeutic decision was apparently appropriate. The presentaiton of the cases is clear and the discussion refers to currently available Guidelines.

Comments:

1. Considering that TBM is mainly a disease of children, why did the authors include only adult patients >18 years? Is this because the hospital only treats adults or is this for demonstrating the difficulty in management of TBM in adults? Or any other reason? I suggest to change the title to "... pathogen-negative tuberculous meningitis in adults: ..." 

2. The insensitivity of classical bacteriological tests (microscopy and culture) is well known and even the new nucleic acid amplification tests (like Xpert MTB/RIF) remain negative in a large proportion of the cases. Therefore, the diagnosis has frequently to rely on clinical, radiological and laboratory informations. In the series presented, all cases had the classical association of high lymphocyte count, high protein and low glucose in CSF, which is very suggestive (although not quite pathognomonic) of TBM. The authors should   give the results of the CSF examination in the result section and refer to Fig 1 (the mention of Fig 1 is missing in the text).

Minor comments:

1. Introduction: "...false negative of M. tuberculosis..."

2. Typing error for TBM (written TMB)

3. Exclusion criteria: I suggest " positive smear" instead of "smear positive"

4. I suggest to replace "rheumatism immunity disease" by "immune-mediated rheumatic disease"

5. Data collection_: please specify that the WCC of CSF and the proportion of different cell types was examined

6. Tab 1: I suggest "history of illness" or "illness history"

7. Tab 1: I suggest to indicate "symptom duration (days)"

8. Results: was the addition of steroids considered (as clearly mentioned in  ref 2 and 4) ?

9. Was rifampicin prescribed in the usual dose (10 mg/kg) or in higher dose (20-30 mg/kg), as suggested in ref 2 and 4?

Author Response

Comments:

1) Considering that TBM is mainly a disease of children, why did the authors include only adult patients >18 years? Is this because the hospital only treats adults or is this for demonstrating the difficulty in management of TBM in adults? Or any other reason? I suggest to change the title to "... pathogen-negative tuberculous meningitis in adults: ..."

Response: The Department of Neurology, Tongji Hospital, Tongji Medical College, Huazhong University of Science and Technology only treats adults, so we did not include children and adolescents under the age of 18, and also changed the title to “Clinical management of pathogen-negative tuberculous meningitis in adults: a series case study”. (page 1, lines 2-3 in Revised-track version)

2) The insensitivity of classical bacteriological tests (microscopy and culture) is well known and even the new nucleic acid amplification tests (like Xpert MTB/RIF) remain negative in a large proportion of the cases. Therefore, the diagnosis has frequently to rely on clinical, radiological and laboratory informations. In the series presented, all cases had the classical association of high lymphocyte count, high protein and low glucose in CSF, which is very suggestive (although not quite pathognomonic) of TBM. The authors should give the results of the CSF examination in the result section and refer to Fig 1 (the mention of Fig 1 is missing in the text).

Response: We have described in the legend of Fig 1 as “See Supplementary Table 1 for details of lumbar puncture time and various index values”. (page 5, lines 29 in Revised-track version)

Minor comments:

1) Introduction: "...false negative of M. tuberculosis..."

Response: The expression of this sentence that we have revised to “The main reasons for a false negative result of M. tuberculosis culture are low abundance of the pathogen in cerebrospinal fluid (CSF), and the need for a high standard of laboratory equipment and procedures.” (page 1, lines 42-44 in Revised-track version)

2) Typing error for TBM (written TMB)

Response: We have corrected all typing errors and revised “TMB” to “TBM”. (page 2, line 4; page 3, line 12)

3) Exclusion criteria: I suggest " positive smear" instead of "smear positive"

Response: We have revised “smear positive” to “positive smear”. (page 2, line 32 in Revised-track version)

4) I suggest to replace "rheumatism immunity disease" by "immune-mediated rheumatic disease"

Response: We have revised “rheumatism immunity disease” to “immune-mediated rheumatic disease”. (page 2, line 37; page 4, line 3-4 in Revised-track version)

5) Data collection: please specify that the WCC of CSF and the proportion of different cell types was examined

Response: The specific values of WCC of CSF and the proportion of different cell types was examined can see in Supplementary Table 1. (page 5, line 29 in Revised-track version)

6) Tab 1: I suggest "history of illness" or "illness history"

Response: We have changed “illness of history” to “illness history”. (page 3, Table 1, line 6; page 4, line 3 in Revised-track version)

7) Tab 1: I suggest to indicate "symptom duration (days)"

Response: We have noted “symptom duration (days)” in Table 1, also noted “ATT time(days)” and “Follow up time(months)”. (page3, Table 1, line 8; page4, Table 1, line 36, line 37 in Revised-track version)

8) Results: was the addition of steroids considered (as clearly mentioned in ref 2 and 4) ?

Response:All patients used dexamethasone in the treatment, which was reflected in the Result chapter in new manuscript. (page 5, line 1 in Revised-track version)

9) Was rifampicin prescribed in the usual dose (10 mg/kg) or in higher dose (20-30 mg/kg), as suggested in ref 2 and 4?

Response:The weight of 5 patients was about 70kg, and they all used 600mg/day rifampicin. Therefore, we use the usual dose (10mg /kg), which is also reflected in our new manuscript. (page 4, line 24-25 in Revised-track version)